# Influence of the SARS-COV2 pandemic on access to healthcare services among people living with HIV and its subsequent effects on antiretroviral therapy uptake in Malawi

Thulani Maphosa[1]*, Malocho Phoso[1], Lucky Makonokaya[1], Louiser Kalitera[1], Rhoderick Machekano[2], Alice Maida[3], Rachel Kanyenda Chamanga[1], Godfrey Woelk[2]

1 Elizabeth Glaser Pediatric AIDS Foundation, Lilongwe, Malawi, 2 Elizabeth Glaser Pediatric AIDS Foundation, Washington, DC, United States of America, 3 U.S. Centers for Disease Control and Prevention, Division of Global HIV and TB, Malawi

* tmaphosa@pedaids.org

## Abstract

The global disruption caused by the SARS-CoV-2 pandemic profoundly affected healthcare systems, particularly impacting People Living with Human Immunodeficiency Virus (PLHIV). This study investigated the repercussions of SARS-CoV-2 infection on access to human immunodeficiency virus (HIV) care and antiretroviral therapy (ARV) in Malawi, emphasizing the critical need to sustain uninterrupted HIV services during health crises. Employing mobile phone-based syndromic surveillance, this study assessed the influence of SARS-CoV-2 on healthcare access for PLHIV across nine districts supported by the Elizabeth Glaser Pediatric AIDS Foundation (EGPAF). Telephone-based interviews were conducted to analyze demographic factors, challenges encountered in accessing HIV services, and adherence to ARV medication, illuminating the pandemic's effects on ARV uptake. The findings revealed that approximately 3.9% (n = 852) of 21981 participants faced obstacles in accessing crucial HIV services during the pandemic, resulting in approximately 1.2% (n = 270) reporting multiple missed doses of ARV medication in a particular month. After adjusting for various variables, males exhibited a higher likelihood of service inaccessibility than females (Adjusted Odds Ratio [AOR] = 1.39, 95% CI: 1.20–1.60, p < 0.001). Age also played a significant role, with individuals aged 35–49 years and those aged 50 years or older demonstrating reduced odds of service failure compared with the reference group aged 18–34 years. Only a small proportion of PLHIV reported disruption in HIV care access, which may be because Malawi did not initiate stringent travel restrictions during the SARS-CoV-2 pandemic. Nonetheless, enduring challenges have been observed in retaining younger PLHIV and men in HIV-care settings. Thus, targeted strategies are imperative for effectively engaging and sustaining these populations in HIV care during and after health crises.

**Data Availability Statement:** All relevant data are within the paper and its Supporting Information files.

**Funding:** The authors declare that this research was supported by President's Emergency Plan for AIDS Relief (PEPFAR) through the Centers for Disease Control and Prevention (CDC) under Cooperative Agreement Number U2UGH002010. The funders had no role in the design of the study, data collection and analysis, decision to publish, or preparation of the manuscript. No additional external funding was received for this study. The authors have no financial relationships or competing interests that could be perceived as influencing the results or interpretation of this research. This surveillance work was supported by the. Its contents are solely the authors' responsibility and do not necessarily represent the official views of PEPFAR, the CDC, or the U.S. Public Health Service. All other authors have declared that no competing interests exist.

**Competing interests:** The authors have declared that no competing interests exist.

## Introduction

The global impact of the severe acute respiratory syndrome coronavirus 2 (SARS-COV-2) has resulted in a multifaceted challenge to the maintenance of health services for individuals with chronic conditions. Movement restrictions, heightened patient anxiety, and strain on resources significantly affect the accessibility and sustainability of healthcare systems [1–3]. Amidst global efforts to contain the SARS-CoV-2 pandemic, a critical concern emerges: millions of individuals living with chronic conditions, including the Human Immunodeficiency Virus (HIV), require uninterrupted and ongoing care [4, 5]. The repercussions of the pandemic extend far beyond the immediate viral threat, posing substantial barriers to the continuum of care for individuals with chronic illnesses such as HIV [6, 7].

Malawi has made significant progress in advancing its healthcare objectives, particularly in the field of HIV treatment. With an estimated 1.1 million individuals living with HIV, the country has made substantial strides in initiating antiretroviral therapy (ARV) in the majority of the affected population [8, 9]. The initiation of ARV signifies a significant milestone in bolstering health care infrastructure, aiming to provide life-saving interventions and sustained care to those grappling with HIV infection. Despite these significant achievements, Malawi has confronted persistent challenges in ensuring uninterrupted treatment for individuals living with HIV, particularly within the complex context of the SARS-CoV-2 pandemic [10]. The pandemic has cast a formidable shadow over healthcare systems worldwide, introducing unprecedented disruptions and amplifying existing vulnerabilities [11].

The confluence of the HIV epidemic and the SARS-CoV-2 pandemic has unveiled a myriad of challenges, intensifying the struggle to maintain consistent access to crucial HIV treatment and care services [12–14]. Interruptions in treatment regimens have emerged as a pressing concern, exacerbated by various pandemic-related factors such as healthcare resource reallocation, logistical hurdles, social restrictions, and patient-specific anxiety [13–15]. Sustaining the continuum of care for individuals living with HIV in Malawi demands innovative approaches and robust strategies that navigate the intricacies of the pandemic while safeguarding the gains achieved in HIV treatment [16]. The critical importance of addressing treatment interruptions (10) and preserving the integrity of HIV care services amidst the ongoing SARS-CoV-2 crisis necessitates a concerted and adaptive response from healthcare systems, policymakers, and communities [5].

In response to the global challenge posed by the SARS-CoV-2 pandemic, various nations, including Malawi, have swiftly implemented preventive measures to mitigate SARS-CoV-2 transmission within ART clinics while maintaining essential services [17]. These measures were comprehensive, encompassing strategies such as augmenting the multi-month dispensation of ART for stable patients and, in some instances, suspending routine clinic visits to minimize physical contact between ART recipients and healthcare providers [9, 17, 18].

As the dual challenge of managing HIV and SARS-CoV-2 persists, it has become increasingly evident that strategies to address these health crises must be holistic, emphasizing the need for resilient healthcare systems capable of concurrently managing multiple health threats. Furthermore, as the pandemic has both exposed weaknesses and unveiled strengths within healthcare systems, there is a pivotal opportunity to fortify these systems, building on strengths while effectively mitigating vulnerabilities to ensure the continuity of essential HIV care within the evolving landscape of the SARS-CoV-2 pandemic [3, 18, 19].

## Methods

### Study design and participants

We conducted a longitudinal serial telephone-based SARS-COV-2 syndromic surveillance survey between July 2021 and April 2022. This study involved People Living with HIV (PLHIV) clients aged $\geq$ 18 years enrolled in ART care across 179 Elizabeth Glaser Pediatric AIDS Foundation (EGPAF)-)-supported facilities in Malawi, southern, and central regions of the country. Study participants were classified according to the region of their current ART facility provider [20].

### Sampling procedure and data collection

Using electronic medical records, client mobile phone numbers were randomly selected and logged into a secure database for enrolment. Trained data collectors, including clinicians and nurses, obtained verbal consent and conducted telephone interviews using structured questionnaires. In order to prevent accidental disclosure of participants' HIV status to other household members during mobile phone interviews, we ascertained the name of the respondent and confirmed if the respondent was in a private space to talk. We also confirmed if they were taking ARVs and from which health facility. The questionnaire (**S1 Text**), which was available in both English and Chichewa, was developed, translated, back-translated, and digitized to ensure consistency and comprehension across languages. Open Data Kit (ODK) software facilitates data capture on secure Android tablets, ensuring real-time consistency checks and storage on a secure online server [20, 21]. Recognizing the stigma associated with COVID-19 which could have impacted the accurate reporting of COVID-19 symptoms by participants, probing was done for specific symptoms.

### Survey variables

The primary outcome variable assessed was the reported incidence of missing any dose of ARVs among PLHIV. The independent variables included sociodemographic factors (sex, education level, age, and regional residence), medical history of SARS-CoV-2-like illness, and history of SARS-CoV-2 symptoms in household members. This study used a standardized set of signs and symptoms of SARS-CoV-2-like illness, including fever, cough, shortness of breath, difficulty breathing, fatigue, muscle or body aches, sore throat, loss of taste or smell, headache, chills, congestion or runny nose, nausea or vomiting, and diarrhea. The demographic variables included sex, age, education level, regional residence, and household and SARS-CoV-2 symptom history, which were strategically selected based on their potential impact on healthcare access and treatment adherence among PLHIV. Similarly, an investigation into the reasons for missing ARVs aimed to uncover the critical factors influencing interruptions in treatment regimens, thereby informing targeted interventions and policy improvements [20] **S1 Text**.

### Sample size determination

We targeted contacts with at least 1,000 PLHIV per week, anticipating a 10% refusal rate. The estimated sample size allowed for the estimation of weekly proportions/rates of the outcomes of interest, ensuring a precision of 2.3% to estimate the prevalence of PLHIV on ART that had challenges in accessing health facilities and missing treatment doses during the study period [20].

### Data management and analysis

Descriptive statistics, including proportions and median estimates, were used for the preliminary data analysis. To explore the association between missing ARV doses and participant characteristics, logistic regression was employed, adjusting for demographic variables and

household experience with SARS-CoV-2-like illness. Logistic regression analysis was performed to identify factors associated with failure to access HIV services in the past month as well as unwillingness to visit a health facility for ARV refill among participants living with HIV [20].

### Ethical considerations

The study was approved by the Malawi National Health Sciences Research Committee and Advarra Institutional Review Board (IRB) (Protocol #20/06/2537). The Centers for Disease Control and Prevention reviewed and approved this project and determined that it would comply with all requirements of surveillance research. Verbal informed consent approved by all the above IRBs, was obtained from all study participants before conducting interviews to ensure that ethical standards were met throughout the study. As part of verification of the participant's name, as recorded in the electronic medical records system, the initiation of the consenting process ensured that the call respondent were the ones registered as HIV positive client enrolled on ART in a specified EGPAF supported facility. Furthermore, the client was asked if he/she is comfortable speaking about his/her health issues at the time of the call.

## Results

From July 2021 to April 2022, the study successfully reached N = 110,402 (76%) active numbers, with N = 44,072 (40%) answering calls. Among those reached, 27,663 (63%) confirmed recorded names, of which 24,416 (88%) were ART clients and 21,981 (94%) were aged 18 or older, and all consented to participate in the study interview. Among the respondents, 54.5% (n = 11,973) were female. The median age of the respondents was 40 years, with 37.8% (n = 8,297) falling within the age bracket 35–44 years. years, and 53.1% had attained secondary education or higher. (**Table 1**).

Geographically, 93.2% of participants were from the southern region. In terms of household health history, close to 7% of the respondents (n = 1,541) reported at least one household member with a history of COVID-19-like illness (CLI), whereas 0.9% (n = 196) reported a household member with a confirmed SARS-CoV-2 infection diagnosis during the study period (**Table 1**).

Approximately 3.9% (n = 852) of the participants faced challenges in accessing essential HIV care services within the preceding month and 1.2% (n = 270) reported missing multiple doses of antiretroviral medications (ARVs), indicating a potential interruption in their treatment regimens (**Table 1**).

We examined the factors contributing to failure to access HIV services within the past month, as presented in **Table 2**. Gender disparities revealed significant associations, with males showing a higher likelihood of failure to access services than females (adjusted odds ratio [AOR] = 1.39, 95% CI: 1.20–1.60, $p < 0.001$). Age was also a contributing factor, with individuals aged 35–49 years (AOR = 0.74, 95% CI: 0.63–0.87, $p < 0.001$) and those aged 50 years or older (AOR = 0.61, 95% CI: 0.49–0.75, $p < 0.001$) demonstrating decreased odds of failing to access HIV services compared to the reference group of PLHIV aged 18–34 years. The educational level and region of respondents who received ARV were not significant determinants.

The analysis presented in **Table 3** evaluates various factors associated with unwillingness to visit a health facility for ARV drug refills among HIV-infected individuals living with HIV. Among the demographic characteristics studied, sex disparities emerged as a notable factor. Males exhibited a higher tendency towards reluctance in seeking facility-based refills compared to females, as evidenced by the AOR of 1.31 (95% CI: 1.22–1.38, $p < 0.001$) after

**Table 1. Distribution of characteristics and HIV care access and ARV dosing of study participants living with HIV and receiving ARV in EGPAF-supported facilities from July 2021 to April 2022 (N = 21 981).**

| Characteristic | Frequency n (%) |
|---|---|
| **Gender** | |
| Female | 11,973 (54.5%) |
| Male | 10,008 (45.5%) |
| **Age range (Years)** | |
| 18–34 years | 6,068 (27.6%) |
| 35–49 years | 11,442 (52.1%) |
| 50+ years | 4,471(20.3%) |
| **Level of Education** | |
| No education | 840 (3.8%) |
| Primary | 9,292 (42.3%) |
| Secondary | 9,980(45.4%) |
| Tertiary | 1,869 (8.5%) |
| **Current Region of Residence** | |
| Northern | 66(0.3%) |
| Central | 1,426 (6.5) |
| Southern | 20,489 (93.2%) |
| **Reported SARS-CoV-2 -like illness in the past two weeks among participants** | |
| Yes | 1,403 (6.4%) |
| No | 20,561 (93.6%) |
| Missing | 17 |
| **Household member history of SARS-CoV-2 -like illness** | |
| Yes | 1,541 (7.0%) |
| No | 20,411 (93.0%) |
| Missing | 21 |
| **Household member history of confirmed SARS-CoV-2 infection** | |
| Yes | 196 (0.9%) |
| No | 21,751 (99.1%) |
| Missing | 34 |
| **Failure to access HIV care by the participant** | |
| Yes | 852 (3.9%) |
| No | 21,081 (96.1%) |
| Missing | 48 |
| **Missed multiple ARV doses in the past month** | |
| Yes | 270 (1.2%) |
| No | 21,660 (98.8%) |
| Missing | 51 |

controlling for other variables. Being aged 50 years or older was associated with lower odds of unwillingness to visit health facilities for ARV refill [(AORs of 0.92 (95% CI: 0.89–0.98, p = 0.019) and 0.90 (95% CI: 0.82–0.98, p = 0.017)], respectively, compared to the reference group of 18–34 years. Participants or household members without a history of SARS-CoV-2-like illness or a confirmed SARS-CoV-2 infection had a lower risk of unwillingness to obtain facility-based ARV refills. Moreover, individuals who reported failure to access HIV care demonstrated a lower likelihood of willingness to obtain facility-based ARV refills (AOR 0.63 (95% CI, 0.55–0.73; p < 0.001).

**Table 2. Factors associated with failure to access HIV services in the past month.**

| Characteristics | Failed to Access HIV Services = Yes n (row %) | Crude Odds Ratio (95% CI) | p-value | Adjusted Odds Ratio (95% CI) | p-value |
|---|---|---|---|---|---|
| **Gender** | | | | | |
| Female | 418 (49.1%) | Ref | | 1 | |
| Male | 434 (50.1%) | 1.25 (1.09–1.44) | 0.001 | 1.39 (1.20–1.60) | <0.001 |
| **Age range** | | | | | |
| 18–34 years | 278 (32.6%) | Ref | | 1 | |
| 35–49 years | 427 (50.1%) | 0.81 (0.69–0.94) | 0.006 | 0.74 (0.63–0.87) | <0.001 |
| > = 50 years | 147 (17.3%) | 0.71(0.58–0.87) | 0.001 | 0.61 (0.49–0.76) | <0.001 |
| **Level of Education** | | | | | |
| No education | 29(3.46%) | Ref | | 1 | |
| Primary | 384(4.14%) | 1.21(0.82–1.77) | 0.34 | 1.17 (0.79–1.72) | 0.42 |
| Secondary | 375(3.77%) | 1.09 (0.74–1.60) | 0.656 | 1.01(0.68–1.48) | 0.97 |
| Tertiary | 64(3.43%) | 0.99(0.63–1.5) | | 0.91 (0.58–1.48) | 0.68 |
| **Current Region of Residence** | | | | | |
| Northern | 5(7.6%) | Ref | | 1 | |
| Central | 64(4.5%) | 0.57(0.22–1.48) | 0.251 | 0.57 (0.22–1.47) | 0.25 |
| Southern | 783(3.83%) | 0.49(0.19–1.21) | 0.122 | 0.48(0.19–1.21) | 0.12 |

## Discussion

From our mobile phone syndromic surveillance, we found that approximately 4% of the participants encountered difficulties in accessing HIV services during the SARS-CoV-2 pandemic. Of these, 15% reported multiple missed doses of ARV. A qualitative study from Zimbabwe described the challenges reported by women on ARVs in accessing health facilities during the COVID-19 restrictions [13]. This study describes how stringent travel restrictions posed formidable barriers to some PLHIV, leading to challenges in attending appointments and collecting vital ARV medications owing to severe constraints on public transport availability [13]. Unlike Zimbabwe and other countries, Malawi adopted a less restrictive approach to preventive interventions during the pandemic, with the Malawi High Court overruling stringent lockdown policies, allowing for continuous public transport, albeit with a reduced seating capacity of 60% [21]. This deviation from the strict measures imposed by other countries possibly contributed to the lower proportion of missing ARV doses and sustained access to healthcare services in our study than in other countries [9, 17, 19]. In contrast to our findings, a study by Cordie in three North African countries found that PLHIV aged ≥60 years had challenges accessing HIV care [19]. This might be due to varying study environments and the maturity of the HIV programme in Malawi where infections are generally higher and huge investments have been made and with most older people being the most experienced in HIV care. We found that younger PLHIV (age 18–34 years) were more likely to report failure to access HIV care services.

Our findings showed that men were more likely to fail to access HIV care and were unwilling to visit health facilities, indicating that they were also more likely to not be retained in care [22–24]. A qualitative study from Malawi reported that men were more inclined to discontinue ART because of travel-related impediments [25]. In this view, there is a need for males to be given special priority in accessing essential healthcare services even during the epidemic period [5, 26].

Moreover, disparities in access to healthcare services across regions became apparent during the pandemic, particularly between rural and urban areas. Studies conducted during this

**Table 3. Factors associated with an unwillingness to visit health facility for ARV refill among participants living with HIV.**

| Characteristics | Not willing to visit the facility for a ARV refill | Crude odds ratio (95% Confidence Interval) | p-value | Adjusted odds ratio (95% Confidence Interval) | p-value |
|---|---|---|---|---|---|
| | n (%) | | | | |
| **Gender** | | | | | |
| Female | 3,109 (26.1%) | Ref | | 1 | |
| Male | 3,103 (31.1%) | 1.28 (1.21–1.36) | <0.001 | 1.29 (1.23–1.39) | <0.001 |
| **Age range** | | | | | |
| 18–34 | 1,721 (28.5%) | Ref | | 1 | |
| 35–49 years | 3,229 (28.3%) | 0.99 (0.93–1.06) | 0.81 | 0.92 (0.87–1.01 | 0.075 |
| > = 50 years | 1,262 (28.3%) | 0.99(0.91–1.08) | 0.85 | 0.89(0.82–0.98) | 0.019 |
| **Level of Education** | | | | | |
| No education | 204(24.3%) | Ref | | 1 | |
| Primary | 2, 653(28.7%) | 1.25 (1.06–1.47) | 0.008 | 1.23(1.04–1.45) | 0.014 |
| Secondary | 2,807 (28.2%) | 1.22(1.04–1.44) | 0.017 | 1.18(1.00–1.39) | 0.046 |
| Tertiary | 548 (29.4%) | 1.3(1.07–1.56) | 0.007 | 1.22(1.02–1.48) | 0.033 |
| **Current region of residence** | | | | | |
| Northern | 10(15.2%) | Ref | | 1 | |
| Central | 402 (28.3%) | 2.21(1.12–4.37) | 0.023 | 2.32(1.17–4.61)) | 0.016 |
| Southern | 5,800(28.4%) | 2.22(1.13–4.35) | 0.02 | 2.38(1.21–4.69) | 0.012 |
| **Participant history of SARS-CoV-2 -like illness** | | | | | |
| Yes | 497 (35.5%) | Ref | | 1 | |
| No | 5,715 (27.9%) | 0.70 (0.63–0.79) | <0.001 | 0.73 (0.65–0.82) | <0.001 |
| **Household member history of SARS-CoV-2 -like illness** | | | | | |
| Yes | 531 (34.5%) | Ref | | 1 | |
| No | 5,681 (27.9%) | 0.73 (0.66–0.82) | <0.001 | 0.81 (0.72–0.90) | <0.001 |
| **Household member history of confirmed SARS-CoV-2 infection** | | | | | |
| Yes | 52 (43.3%) | Ref | | 1 | |
| No | 340 (35.5%) | 0.60 (0.45–0.79) | <0.001 | 0.65 (0.48–0.87) | 0.004 |
| **Failure to access HIV care by the participant** | | | | | |
| Yes | 334 (39.2%) | Ref | | 1 | |
| No | 5,878 (27.9%) | 0.60 (0.52–0.69) | <0.001 | 0.63 (0.55–0.73) | <0.001 |

period highlighted a significant increase in missed visits among PLHIV in rural settings compared to their urban counterparts [27]. This discrepancy is attributed to the allocation of more resources in urban settings, potentially rendering rural areas more vulnerable to service disruption. While our study did not explicitly explore this regional variable, acknowledging and understanding such regional variations is pivotal for formulating targeted interventions to bridge gaps in healthcare access. Recognizing these disparities enables the development of context-specific strategies tailored to address the unique challenges faced by rural communities, thus ensuring equitable access to healthcare services for all individuals regardless of geographic location [21]. The overwhelming concentration of participants hailing from the southern region of Malawi from our study is expected. However, this geographical bias is not merely a coincidence but a consequence of the deliberate focus on EGPAF-supported sites, predominantly situated in the south. Moreover, the dynamic nature of human mobility in Malawi further complicates the picture. Despite our best efforts, the fluidity of client movement across regions introduces an element of uncertainty. It's entirely plausible that some study subjects, initially hailing from the southern sites, have since migrated to the northern reaches of the country [21].

Our study had some limitations. As our investigation was conducted via phone-based interviews, our study inherently excluded individuals without access to mobile phones, potentially omitting a substantial proportion of the population estimated to be approximately 50% of households in the country. This exclusion affects the generalizability of the results to the overall population. Moreover, reliance on self-reported data introduces potential biases, including recall and social desirability biases, which might have influenced participants' responses. Additionally, our study's restricted scope to specific EGPAF-supported districts limits its broader generalizability, emphasizing the need for caution when extrapolating the findings to the entire Malawian PLHIV population [20]. Moreover, our study could not evaluate the correlation between duration on ART and client age, a crucial determinant potentially influencing adherence to scheduled clinic appointments. This oversight regarding a pivotal variable underscores a missed opportunity to unveil nuanced insights into treatment compliance dynamics. Addressing this gap could illuminate critical pathways for optimizing patient care strategies and enhancing overall health outcomes.

## Conclusion

Our study found that a relatively small proportion of PLHIV did not access HIV care services, probably because of the government's decision to not implement stringent SARS-CoV-2 pandemic restrictions, such as limiting travel. However, the groups that were more likely to report not accessing services and unwilling to travel to health centers were younger PLHIV and men, reflecting the long-standing challenges of maintaining these populations in HIV care. This emphasizes the urgent need to implement effective strategies to engage men and youth to be retained in care.

## Supporting information

**S1 Text. PLHIV COVID 19 syndromic surveillance questionnaire.**
(DOCX)

## Acknowledgments

The authors are grateful to the Government of Malawi for providing a platform for this syndromic surveillance system. We also gratefully acknowledge the support from the Malawi Ministry of Health Department of HIV/AIDS, the PEPFAR/CDC team in Malawi and Atlanta, and EGPAF Malawi for their technical role and leadership in establishing the syndromic surveillance system.

## Author Contributions

**Conceptualization:** Thulani Maphosa, Malocho Phoso, Lucky Makonokaya, Rhoderick Machekano, Alice Maida, Rachel Kanyenda Chamanga, Godfrey Woelk.

**Data curation:** Lucky Makonokaya, Louiser Kalitera, Rhoderick Machekano, Godfrey Woelk.

**Formal analysis:** Thulani Maphosa, Lucky Makonokaya, Louiser Kalitera, Rhoderick Machekano.

**Funding acquisition:** Thulani Maphosa.

**Investigation:** Thulani Maphosa, Rhoderick Machekano, Rachel Kanyenda Chamanga.

**Methodology:** Rhoderick Machekano, Alice Maida, Godfrey Woelk.

**Project administration:** Thulani Maphosa, Rachel Kanyenda Chamanga, Godfrey Woelk.

**Supervision:** Thulani Maphosa.

**Validation:** Alice Maida, Godfrey Woelk.

**Writing – original draft:** Thulani Maphosa.

**Writing – review & editing:** Thulani Maphosa, Malocho Phoso, Lucky Makonokaya, Alice Maida, Godfrey Woelk.

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
