## [Decision Letter · Decision Letter 0]

1 May 2024

PGPH-D-24-00421

Influence of the SARS-COV2 Pandemic on Access to Healthcare Services Among People Living with HIV and its Subsequent Effects on Antiretroviral Therapy Uptake in Malawi.

Dear Dr. Maphosa,

Thank you for submitting your manuscript to PLOS Global Public Health. After careful consideration, we feel that it has merit but does not fully meet PLOS Global Public Health’s publication criteria as it currently stands. Therefore, we invite you to submit a revised version of the manuscript that addresses the points raised during the review process.

Please address feedback from reviewers 1 and 2 and resubmit.

We look forward to receiving your revised manuscript.

Kind regards,

Heather Haq, M.D., M.H.S.

Academic Editor

Journal Requirements:

2. In the ethics statement in the Methods, you have specified that verbal consent was obtained. Please provide additional details regarding how this consent was documented and witnessed, and state whether this was approved by the IRB.

Additional Editor Comments (if provided):

Reviewers' comments:

Reviewer's Responses to Questions

**Comments to the Author**

1. Does this manuscript meet PLOS Global Public Health’s publication criteria? Is the manuscript technically sound, and do the data support the conclusions? The manuscript must describe methodologically and ethically rigorous research with conclusions that are appropriately drawn based on the data presented.

Reviewer #1: Yes

Reviewer #2: Yes

2. Has the statistical analysis been performed appropriately and rigorously?

Reviewer #1: Yes

Reviewer #2: Yes

3. Have the authors made all data underlying the findings in their manuscript fully available (please refer to the Data Availability Statement at the start of the manuscript PDF file)?

Reviewer #1: Yes

Reviewer #2: Yes

4. Is the manuscript presented in an intelligible fashion and written in standard English?

Reviewer #1: Yes

Reviewer #2: Yes

5. Review Comments to the Author

Reviewer #1: In reviewing the manuscript titled "Influence of the SARS-COV2 Pandemic on Access to Healthcare Services Among People Living with HIV and its Subsequent Effects on Antiretroviral Therapy Uptake in Malawi," I commend the authors for their meticulous approach and understanding of the multifaceted challenges faced by People Living with HIV (PLHIV) during the COVID-19 pandemic. This article is well-conceptualized and integrates epidemiological data with qualitative assessments, analyzing the pandemic's repercussions on this vulnerable population's access to healthcare services.

The manuscript highlights the complex interplay between the SARS-CoV-2 outbreak and healthcare accessibility for PLHIV, underlining critical areas such as delays in accessing antiretroviral therapy, disruptions in HIV service delivery, and the impact of the pandemic on retention in care for PLHIV.

Furthermore, the study's conceptualization incorporates the unique challenges posed by a global pandemic. The findings allow an understanding of the barriers to healthcare access, emphasizing the significance addressing the impact of social determinants of health on continuity of care. This manuscript is a commendable contribution to literature, offering valuable insights into the impact of the SARS-CoV-2 pandemic on access to healthcare services among PLHIV.

I would like the author to elaborate on the following aspects of the manuscript:

1) What measures did the researchers implement to prevent accidental disclosure of participants' HIV status to other household members during mobile phone interviews?

2) Given the stigma that was associated with COVID-19 during the early onset of the pandemic in Africa, how did the study implementation team ensure accurate reporting of COVID-19 symptoms by participants?

3) Was treatment experience, as opposed to client's age, a confounding or determining variable for better compliance to scheduled Clinic appointments?

4) What factors led to most participants being those that resided in the south of Malawi? Is it an urban region of the country?

Reviewer #2: This study is a longitudinal, cross-sectional study conducted via telephone interview of PLHIV in Malawi during a nine-month period from 2021-2022. The study sought to determine whether PLHIV and receiving care at EGPAF-supported clinics encountered barriers to care during the COVID pandemic. Survey answers were then analyzed using first descriptive statistics, then logistic regression analysis to determine if any discrepancies seen were statistically significant. 3.9% of patients interviewed experienced challenges in accessing essential HIV services during the prior month and only 1.2% missed doses of ART, with the authors hypothesizing that these relatively low numbers might be related to the lack of restrictions on public transportation during the COVID pandemic in Malawi. Additionally, of the respondents, both men and clients aged 18-34 years were found to be the most likely to experience interruptions in or barriers to care during this timeframe, which is in line with the majority of HIV access-to-care research. The study had some clear limitations, the most obvious of which was exclusion of patients without access to mobile phones, as the authors conducted a telephone-based survey. This accounts for approximately 50% of households in Malawi, which is a significant proportion. Likewise, the limitation of the study to EGPAF-funded districts limits some generalizability, as does the study being predominantly performed in the southern region of Malawi.

Overall, this was a well done study analyzing how the COVID pandemic affected access to care in primarily the southern districts of Malawi. The introduction was very strongly written, highlighting the need for such studies and the role the findings would play in understanding barriers to care for patients. It confirmed some of the already known information about at-risk populations of PLHIV (males, in particular), and highlighted that the surveyed areas of Malawi experienced fewer disruptions due to COVID than other subsaharan African countries. However, the authors did briefly mention that there was a referenced study showing that clients older than 60 tended to struggle to access HIV care in North African countries, but then did not include further discussion on why this discrepancy might have been present. I feel it tends to be more common for younger populations to struggle with accessing HIV care (though often these populations are younger than 18 years old), so it would be interesting to see why the authors included/referenced a study in the discussion that contradicted their findings without further exploration.

In looking at the telephone survey, even though the relative number of clients experiencing disruptions was low, it would have been interesting to see a discussion/analysis of the reasons for disruptions, as this was asked in the client survey.

The results in tables 2 and 3 were somewhat confusing; it seems that one examined patients who experienced "failure to access HIV services" while the other included those with "unwillingness to visit a health facility." In particular, I found the sentence "Moreover, individuals who reported no failure to access HIV care demonstrated a lower likelihood of unwillingness to obtain facility-based ARV refills (AOR 0.63 (95% CI, 0.55-0.73; p < 0.001)." somewhat confusing. The authors may want to elucidate a bit more clearly what exactly they are trying to show by highlighting these two populations of survey respondents.

Additional comments:

- in the introduction section of abstract, antiretroviral therapy (ART) is introduced, and then ARV is the acronym used in the results; would recommend choosing either ART or ARV for the entire paper

6. PLOS authors have the option to publish the peer review history of their article (what does this mean?). If published, this will include your full peer review and any attached files.

**Do you want your identity to be public for this peer review?** For information about this choice, including consent withdrawal, please see our Privacy Policy.

Reviewer #1: **Yes: **Dr. Florence Anabwani-Richter

Reviewer #2: No

---

## [Decision Letter · Decision Letter 1]

9 Aug 2024

Influence of the SARS-COV2 Pandemic on Access to Healthcare Services Among People Living with HIV and its Subsequent Effects on Antiretroviral Therapy Uptake in Malawi.

PGPH-D-24-00421R1

Dear Dr Maphosa,

We are pleased to inform you that your manuscript 'Influence of the SARS-COV2 Pandemic on Access to Healthcare Services Among People Living with HIV and its Subsequent Effects on Antiretroviral Therapy Uptake in Malawi.' has been provisionally accepted for publication in PLOS Global Public Health.

Best regards,

Julia Robinson

Executive Editor

Reviewer Comments (if any, and for reference):

Reviewer's Responses to Questions

**Comments to the Author**

1. If the authors have adequately addressed your comments raised in a previous round of review and you feel that this manuscript is now acceptable for publication, you may indicate that here to bypass the “Comments to the Author” section, enter your conflict of interest statement in the “Confidential to Editor” section, and submit your "Accept" recommendation.

Reviewer #1: All comments have been addressed

2. Does this manuscript meet PLOS Global Public Health’s publication criteria? Is the manuscript technically sound, and do the data support the conclusions? The manuscript must describe methodologically and ethically rigorous research with conclusions that are appropriately drawn based on the data presented.

Reviewer #1: Yes

3. Has the statistical analysis been performed appropriately and rigorously?

Reviewer #1: Yes

4. Have the authors made all data underlying the findings in their manuscript fully available (please refer to the Data Availability Statement at the start of the manuscript PDF file)?

Reviewer #1: Yes

5. Is the manuscript presented in an intelligible fashion and written in standard English?

Reviewer #1: Yes

6. Review Comments to the Author

Reviewer #1: (No Response)

7. PLOS authors have the option to publish the peer review history of their article (what does this mean?). If published, this will include your full peer review and any attached files.

**Do you want your identity to be public for this peer review?** For information about this choice, including consent withdrawal, please see our Privacy Policy.

Reviewer #1: **Yes: **Dr. Florence Anabwani-Richter
